# Comparative Evaluation of Ultrasound Measurement of the Plantar Fascia Between Expert and Novice Technicians

**DOI:** 10.3390/healthcare13192484

**Published:** 2025-09-30

**Authors:** Alba Larriba-Pérez, Mª Carmen Ledesma-Alcázar, María Teresa García-Martínez, Carmen García-Gomariz, José-María Blasco, Paula Cobos-Moreno

**Affiliations:** 1Department of Nursing, University Center of Plasencia, University of Extremadura, 10600 Cáceres, Spain; alarriba@alumnos.unex.es (A.L.-P.); pacobosm@unex.es (P.C.-M.); 2Department of Anatomy, Cell Biology and Zoology, University of Extremadura, 10600 Cáceres, Spain; mledesma@unex.es; 3Ankle and Foot Reserach Advances Group—GAITP, Department of Nursing, University of Valencia, Av. Menéndez y Pelayo 19, 46010 Valencia, Spain; maria.t.garcia-martinez@uv.es; 4Group of Physiotherapy in the Ageing Process: Social and Health Care Strategies, Department of Physiotherapy, University of Valencia, Gascó Oliag 5, 46010 Valencia, Spain; jose.maria.blasco@uv.es; 5Department of Physiotherapy, University of Valencia, Gascó Oliag 5, 46010 Valencia, Spain

**Keywords:** plantar fascia, plantar fasciitis, ultrasound, expert technician, novice technician

## Abstract

**Background and Objectives**: Plantar fasciitis is one of the most common foot pathologies, and its diagnosis and clinical follow-up increasingly rely on the use of ultrasound. The aim of this study is to compare the reliability of ultrasound measurements of plantar fascia thickness between an expert technician, with more than 5 years of ultrasound experience, and a novice technician, with no prior ultrasound experience—both of whom are podiatrists. This allows us to assess whether operator experience significantly influences the results. **Methods**: An observational, descriptive, and cross-sectional study was designed with a sample of 60 healthy patients aged between 20 and 32 years. The thickness of the plantar fascia in both feet was measured using ultrasound. Each patient was evaluated by two observers (one expert and one novice) using a Vinno E35 ultrasound machine. **Results**: The results of the analysis indicated that there were no statistically significant differences in the measurements obtained either between the two technicians or between the left and right feet of the same individual, as the calculated *p*-value in both cases was greater than the conventional threshold of 0.05. This suggests that the measurements were consistent regardless of the operator or the side being evaluated. Nevertheless, when examining the differences in the time required to measure the plantar fascia between the two technicians, the situation was different. In this case, the data distribution did not meet the assumption of normality, as evidenced (*p*-value of less than 0.001). Furthermore, it was observed that the experienced technician not only completed the measurements in a shorter amount of time but also demonstrated less variability in those times, indicating a more efficient and standardized approach to the procedure. In contrast, the novice technician initially took longer and exhibited greater inconsistency; however, as the study progressed, a noticeable and progressive learning effect became apparent. Specifically, from approximately the midpoint of the study onward, the novice technician showed a significant improvement, achieving faster and more consistent measurement times compared to the earlier stages of the research. **Conclusions**: The study demonstrates that ultrasound measurements of the plantar fascia are consistent between technicians. However, the expert technician performs the measurements with greater speed and precision, and a learning effect is evident in the novice technician.

## 1. Introduction

The plantar fascia is an important anatomical structure located on the sole of the foot, playing a fundamental role in maintaining proper biomechanics of the lower limb. It is a connective, fibrous, and dense regular tissue that extends from the heel to the toes, forming a key element of the plantar support system. Its main functions are to support the foot arch, absorb impact forces generated during walking, running, or prolonged standing, and to contribute to the control of both foot and ankle movement [1,2].

In this sense, the plantar fascia acts not only as a passive structure but also as a dynamic stabilizer that helps distribute mechanical loads across the foot. The central component of the plantar fascia is the most robust, longest, and strongest part. It is located in relation to the flexor digitorum brevis muscle and originates from the medial plantar tubercle of the calcaneal tuberosity. From this point, the fascia extends anteriorly in a fan-like arrangement until it reaches the metatarsal heads, where it divides into superficial and deep portions [3]. This unique structural organization provides both elasticity and resistance, enabling the fascia to sustain repetitive stresses over time [4].

One of the most common and clinically relevant pathologies associated with this structure is plantar fasciitis, which refers to the inflammation and degeneration of the fascia, often resulting from mechanical overload or extended periods of standing [3]. This condition typically manifests as intense heel pain, particularly noticeable during the first steps in the morning, when the fascia is stretched after a period of rest. Patients suffering from plantar fasciitis usually report localized pain at the calcaneal tubercle, which can significantly impair mobility and quality of life. Although the precise etiology remains unclear, it is generally accepted that the condition is multifactorial, with abnormal biomechanics, delayed tissue healing, and repetitive microtrauma being potential contributors [4]. Plantar fasciitis may affect both athletic and non-athletic populations. It represents the most common cause of musculoskeletal pain in the foot and is reported in approximately 8% to 25% of injured runners and athletes [4].

Nevertheless, it is not limited to sports-related activities, as it is also commonly observed in individuals with occupations requiring long hours of performing standing, walking, or weight-bearing tasks. Several risk factors have been identified, with limited ankle dorsiflexion being one of the most prominent. This limitation in joint mobility restricts the range of motion, thereby generating compensatory forced plantar flexion during gait. Furthermore, a body mass index (BMI) greater than 30 kg/m^2^ has been strongly associated with an increased risk of plantar fasciitis, as excess body weight imposes additional mechanical stress on the fascia. Occupational factors, particularly those involving prolonged standing or weight-bearing on one foot, are also considered significant contributors. These findings highlight the importance of maintaining a healthy lifestyle, managing body weight, and incorporating targeted ankle and foot exercises. Preventive strategies focused on these modifiable risk factors may reduce the likelihood of developing plantar fasciitis and, in cases where it occurs, may attenuate symptom severity [5,6].

The diagnosis of plantar fasciitis is generally clinical and, in most cases, does not require advanced imaging. However, complementary techniques such as ultrasound can support diagnostic accuracy, especially when plantar fascia thickening below 4 mm is observed, accompanied by hypoechoic changes. Despite being a frequent condition, the clinical course is not completely defined. Evidence suggests that approximately one in ten cases eventually requires surgical intervention, whereas the remaining majority resolve with conservative treatment within 12 months [7].

Treatment strategies may vary depending on the stage and severity of the pathology. In the acute phase, corticosteroid injections are sometimes used. These anti-inflammatory agents inhibit the release of pro-inflammatory substances within the fascia, offering temporary relief. Nevertheless, their use should be carefully restricted to a maximum of three injections per year, as repeated applications carry the risk of tissue crystallization and potential rupture. In addition, other injectable therapies have been investigated, including Platelet-Rich Plasma (PRP), which is rich in growth factors that stimulate tissue repair, and collagen-based injections designed to promote structural regeneration [8].

Conservative management remains the cornerstone of treatment. Shockwave therapy, which applies microsecond-duration acoustic pressure impulses ranging from 35 to 120 MPa, has been shown to stimulate local tissue healing by promoting better alignment of collagen fibers. Similarly, the prescription of custom plantar orthoses plays a central role in addressing biomechanical alterations. These orthotic devices often include cushioning elements in the heel or compensatory adjustments for posterior muscle retraction, thereby improving shock absorption and reducing excessive tensile forces on the fascia [2].

Ultrasound has gained growing relevance as both a diagnostic and therapeutic follow-up tool in plantar fasciitis. Beyond confirming the diagnosis, ultrasound imaging provides a precise and non-invasive method to measure plantar fascia thickness and detect pathological variations. Moreover, it serves as a valuable resource for monitoring treatment progress and guiding the clinical decision-making process, particularly when considering whether to persist with conservative management or escalate to alternative therapeutic options [9].

The rationale for conducting the present study arises from the increasing importance of ultrasound in podiatric practice, coupled with the need to ensure accurate interpretation of plantar fascia measurements. Of particular concern is the influence of the examiner’s level of experience on the reliability of the measurements, which may potentially compromise clinical follow-up. For this reason, it is especially important that the recorded values be reproducible and independent of the technician’s expertise, thus guaranteeing consistency and reliability in patient management.

Accordingly, the main objective of this study is to measure the thickness of the plantar fascia using ultrasound and to compare the results obtained by a highly experienced technician, with more than five years of expertise, against those measured by a novice examiner with no prior exposure to the technique. By evaluating a population of healthy patients, the aim is to determine whether significant inter-individual differences arise between the two sets of measurements, thereby assessing the reproducibility and reliability of this diagnostic approach in clinical practice.

## 2. Methods

### 2.1. Participants

The study sample will consist of a total of 60 patients attending the Podiatric Clinic of the University of Extremadura, aged between 20 and 32 years, including both men and women (Table 1).

Based on the study’s inclusion criteria, a sample of patients of both sexes aged between 18 and 32 years will be selected. The inclusion will consider both patients who attend the clinic due to heel pain and those who do not report such discomfort. Selection will be made randomly within the age range defined for the study. On the other hand, the exclusion criteria will rule out individuals who are unable to walk independently or who require external assistance to do so, as well as those diagnosed with neurodegenerative diseases, in order to avoid biases that could affect the reliability of the analysis results.

To avoid comparisons, the novice technician will perform the measurements first, and will be alone in the examination room with the patient. The novice technician will then leave the examination room and the experienced technician will enter. The technician will follow the same procedure. Each technician will take three measurements on each foot.

The procedure will begin with the patient lying supine on the examination table. The plantar fascia insertion area, i.e., the area of interest for the corresponding measurements, will be palpated. Once the area is located, we will apply conductive gel to the transducer and, positioning it longitudinally, we will search for the fascia insertion area.

When we have located the desired area, we will freeze the image to perform the measurements more accurately. Using the distance program, we will mark the three measurements for each foot. The Bioethics and Biosafety Committee (Registration No.: 53//2025) of the University of Extremadura has reviewed and approved the evaluation protocols and data handling procedures included in this research. All participants took part in the study voluntarily, were informed of both the purpose and the usefulness of the requested data and signed an informed consent form.

### 2.2. Data Collection

Measurements are carried out on a cohort of healthy subjects in order to determine whether there are significant differences between the two observers. Data will be collected using an Excel spreadsheet. All patients will be identified by a previously assigned numerical code. The data collection sheet will include the following parameters of interest: patient number, sex, age, weight, height, pain, whether they engage in sports, the type of sport practiced, and the measured thickness of the fascia.

### 2.3. Procedures

For sample collection, we will use the Vinno E35 ultrasound machine, VINNO, Suzhou, China (Figure 1) with a linear transducer (6–12 MHz) (Figure 2).

Initially, the patient’s identification data—including full name and both surnames—are entered into the ultrasound system software to ensure accurate and organized record-keeping. This step is essential for maintaining traceability of the measurements and preventing any potential confusion during subsequent data analysis.

To minimize potential bias and enhance the reliability of the results, each observer performs three independent measurements per foot. This approach improves the precision, consistency, and reproducibility of the data, thereby increasing the overall validity of the study and reducing the likelihood of measurement errors or sample distortion.

Both feet of each patient are evaluated to obtain a comprehensive assessment. The procedure follows a structured protocol: first, a novice technician conducts the initial measurements while alone with the patient in the examination room to avoid external influences. Upon completion, an experienced technician enters the room and repeats the exact same protocol to ensure methodological consistency across observers.

With the patient comfortably positioned in a supine posture on the examination table, the anatomical location of interest is identified through manual palpation. The primary target area is the insertion point of the plantar fascia, which serves as the key region for ultrasound evaluation. Once this area is located, an appropriate amount of conductive gel is applied to the transducer to ensure optimal transmission of ultrasound waves and high-quality imaging.

The transducer is then positioned longitudinally relative to the foot, and a precise ultrasound scan is performed to locate the insertion point of the plantar fascia. Once clearly visualized, the image is frozen on the ultrasound system to allow for accurate measurements. Using the “distance” measurement tool available in the software, three measurements are taken per foot: one at the exact insertion point of the fascia, a second at the mid-body of the fascia, and a third at a more distal location.

After all measurements are recorded and properly stored within the ultrasound software, the corresponding images are saved to ensure that all data are available for subsequent analysis and comparison (Figure 3).

### 2.4. Data Analysis

The inferential statistical tools used in the present study primarily involved non-parametric tests [8], as many of the quantitative variables analyzed did not follow a normal distribution, according to the results of the Kolmogorov–Smirnov and Shapiro–Wilk normality tests. Consequently, the Wilcoxon signed-rank test was used to compare repeated measurements within the same group. The selection of these tests enables a robust analysis of the differences observed between technicians, as well as under different study conditions, while maintaining the validity of the results despite the lack of normality in several distributions.

The data were processed using IBM SPSS Statistics version 21 for iOS, the official software licensed by the University of Extremadura (see https://arquimedes.unex.es/).

## 3. Results

In the evaluation of variations in ultrasound measurements, we found no significant difference between the measurements of the right and left foot for either technician (expert or novice). Therefore, both feet behave similarly (*p*-value > 0.05) (see Table 2). The descriptive statistics table allows for the examination of variability in the ultrasound measurements, and for this reason, the data have been grouped according to foot (RF: Right Foot; LF: Left Foot) and technician (UT: Untrained = Novice; T: Trained = Expert). In addition, a mean was calculated for each data set (MEAN_RF_UT, MEAN_LF_UT), which summarizes the overall measurement patterns in each condition.

In the comparative analysis of the differences in measurement time between an expert technician and a novice technician, it was identified that the differences did not follow a normal distribution (*p* < 0.001; Table 3). There was a measurement difference between both technicians with a *p*-value < 0.001, and a Cohen effect of 0.81). The expert technician showed more efficient and consistent performance, characterized by rapid and accurate identification of the anatomical structures. In contrast, the novice technician presented greater variability and longer measurement times, attributable to less experience with the technique. These differences reflect interobserver variability related to the level of training, but do not constitute a systematic bias that compromises the validity of the measurement.

Finally, to evaluate the statistical significance of the improvement in measurement time by the novice technician, the Wilcoxon signed-rank test was used, yielding a *p*-value < 0.001 (see Table 4), indicating a statistically significant difference. This was demonstrated by dividing the observed individuals into two phases: the first phase consisted of the first 30 individuals, and the second phase of the remaining 30. When comparing these two phases, a very significant improvement in measurement times was observed in the second phase, where times were shorter, with a Cohen effect of 0.62. The study evidences a substantial improvement in the novice technician’s measurement times.

## 4. Discussion

This study assesses how measurements performed by the expert technician exhibited notable consistency, with reduced variability and significantly shorter execution times. This efficiency is attributed to the technician’s extensive experience and familiarity with ultrasound techniques, which facilitates rapid identification of anatomical landmarks and precise measurement execution. The scientific literature supports this correlation between experience and reliability, emphasizing that both prior training and accumulated practice positively influence the accuracy and speed of ultrasound assessments [10,11,12].

In contrast, the novice technician required longer measurement times and showed greater variability, particularly during the initial assessments. This discrepancy can be explained by limited experience and reduced familiarity with ultrasound imaging and the anatomical visualization of the plantar fascia. These factors may lead to slower identification of target structures and necessitate repeated measurements to ensure accuracy. Previous studies have shown that while novice operators can achieve acceptable levels of reliability after structured training, they initially exhibit significant variability in the time required to complete measurements [13].

The direct comparison between both technicians highlights a statistically significant difference in measurement times, underscoring the importance of accumulated experience in procedural efficiency, as noted by G. Crofts in his research [11]. However, it is important to emphasize that novice technicians can progressively improve their performance and reduce temporal variability as they gain experience and consolidate their skills, a trend supported by complementary studies [13].

One of the most noteworthy aspects of this study is the observation of a progressive improvement in measurement times by the novice technician. This trend becomes particularly evident when the evaluated subjects are divided into two phases: the first phase comprising the initial 30 individuals and the second phase including the remaining 30. A significant improvement in measurement times was observed during the second phase. Specifically, in the first phase, times decreased from 4 min and 37 s to 2 min and 30 s, while in the second phase, the reduction was even more pronounced, from 1 min and 33 s to 1 min and 15 s. This improvement was statistically confirmed by a highly significant Wilcoxon test result (*p* < 0.001), indicating a clear learning effect throughout the study, and an effect size with a Cohen effect of 0.62. This methodological strategy reduces the heterogeneity attributable to the level of expertise of the evaluator, a factor recognized as influential in the reliability and consistency of measurements. Comparing the results between these two phases provides a clearer estimate of the effect size, which makes it easier to identify differences attributable to professional experience and, consequently, more robustly interpret the practical relevance of the findings [14].

The results obtained highlight the influence of the learning curve on ultrasound measurement of the plantar fascia. The fact that the first 30 measurements showed greater variability and less consistency compared to the last 30 supports the hypothesis that experience is a determining factor in the quality of the ultrasound recording. The progressive repetition of the technique contributes to improved accuracy, reduced intra-observer variability, and increased examiner confidence. Thus, the observed evolution not only reflects greater technical skill but also that experience influences the quality and consistency of measurement.

This phenomenon is not isolated; similar patterns have been documented in related contexts. For example, Ámbar J. Vogt reported that novice operators experience a marked learning curve during initial ultrasound sessions, progressively improving both speed and accuracy of measurements [15]. Likewise, a study conducted by David P. Bahner on clinical ultrasound training demonstrated that acceptable levels of competence can be achieved in relatively short periods through brief, structured training and systematic repetition of the technique [15,16]. These findings reinforce the idea that while prior education is a critical factor, continuous practice is equally essential for skill consolidation. The evidence gathered in this study may serve as a foundation for designing future training programs that incorporate progressive and supervised learning, aiming to reduce the initial gap between experienced and novice technicians and ensure greater uniformity in the quality and efficiency of ultrasound measurements [15].

Young Hwan Park et al. have demonstrated that plantar fascia measurement is highly reliable when performed by expert technicians. They also recommend averaging multiple measurements and using standardized protocols to optimize reliability [17]. This supports the need for structured training and protocol standardization before using measurements in a comparative study. In our case, the novice technician underwent three sessions of three hours each related to theoretical training on the ultrasound anatomy of the plantar fascia and common artifacts, technical parameters and common errors (angle, pressure, zoom, focus), and image quality indicators and how to decide whether to repeat the measurement [18].

In another way this study presents a comprehensive analysis of ultrasound measurements performed on both feet by technicians with varying levels of experience, offering valuable insights into the reliability and consistency of these measurements regardless of the operator’s expertise. This aspect is particularly relevant in clinical and research settings, where objectivity in measurement is essential for establishing diagnostic criteria and conducting comparative evaluations.

The findings of this investigation clearly demonstrate that there are no statistically significant differences between measurements taken on the right and left foot, whether performed by an experienced technician or a novice. This suggests a bilateral symmetry in plantar fascia thickness among healthy individuals, a result that aligns with previous research. For instance, Ribeiro et al. reported that in asymptomatic subjects, no clinically relevant differences were observed in fascial thickness between both feet of the same individual [9]. These data reinforce the notion that laterality is not a determining factor in the ultrasound assessment of the plantar fascia.

Moreover, the statistical analysis conducted in this study supports the conclusion that the absence of significant differences between measurements taken by technicians with different levels of experience indicates that intra-individual laterality does not constitute a critical variable when evaluating fascial thickness via ultrasound. This finding strengthens the position of ultrasound as a reliable and objective imaging modality for comparative studies. In fact, this conclusion is consistent with the work of Mohseni-Bandpei, who demonstrated that ultrasound is a highly dependable technique not only for assessing plantar fascia thickness but also for monitoring therapeutic interventions and guiding specific clinical procedures [9].

Comparisons with patients with plantar fasciitis will require confirmation through specific validation studies in symptomatic patients. This is a distinct population where tissue structure and properties (thickness, echogenicity, fibrosis) differ between healthy and diseased patients, which can affect the performance of the ultrasound technician and operator. Furthermore, increased variability may be observed in the presence of heterogeneity, as focal lesions or edema often increase measurement variability and reduce the reliability between both observed.

The limitations of this research include a significant obstacle, such as the limited prior literature, which complicates external validation and comparison of the research findings. However, we conclude that the research’s strengths include the significant clinical impact of ultrasound in podiatry, and the innovative potential for future research, opening up new lines of research.

## 5. Conclusions

This study highlights the robustness and reliability of ultrasound measurements of the plantar fascia when performed under standardized protocols, regardless of the technician’s level of experience. Following a thorough analysis of the measurements obtained from both feet and by different operators, the results indicate that there are no statistically significant differences in the values recorded between the participating professionals. Similarly, no relevant discrepancies were observed between the measurements of the right and left foot, reinforcing the consistency and symmetry of the procedure when executed correctly.

Regarding the time required to complete the measurements, the data show that the experienced technician performed the assessments significantly faster and with greater consistency. This efficiency is attributed to the technician’s advanced familiarity with the ultrasound equipment and their refined ability to accurately identify anatomical structures, such as the plantar fascia. These skills contribute to minimizing measurement errors and reducing variability, thereby enhancing the overall reliability of the data collected.

In contrast, the novice technician initially exhibited longer execution times and greater variability in the measurements. This difference is directly related to their limited experience and the inherent challenges in rapidly and accurately identifying anatomical landmarks. These findings underscore the importance of structured training and systematic practice in the development of ultrasound competencies.

Importantly, the study also reveals a clear learning effect in the less experienced technician, evidenced by a progressive and substantial improvement in measurement times as the study progressed. This trend demonstrates the positive impact of continuous training and hands-on experience on the standardization and reliability of ultrasound assessments. Over time, the novice technician was able to reduce execution times and improve consistency, approaching the performance levels of the experienced operator.

This observation reinforces the value of ultrasound as a dependable diagnostic tool for evaluating the plantar fascia, independent of the technician’s experience level. While experience influences the speed of measurement, it does not significantly affect the accuracy of the results when appropriate protocols are followed. These findings support the implementation of structured training programs that emphasize progressive skill acquisition and supervised practice, aiming to reduce initial disparities between novice and expert technicians. Such programs would contribute to greater uniformity in the quality and efficiency of ultrasound measurements, ultimately enhancing diagnostic precision and clinical decision-making.

Therefore, it can be concluded that a novice technician may need to analyze at least more than 60 samples before achieving a sufficient level of skill to ensure consistent and clinically reliable ultrasound measurements of the plantar fascia. This finding underscores the importance of including a supervised training period with an adequate number of examinations before considering the technique fully valid in clinical practice and research.

## Figures and Tables

**Figure 1 healthcare-13-02484-f001:**
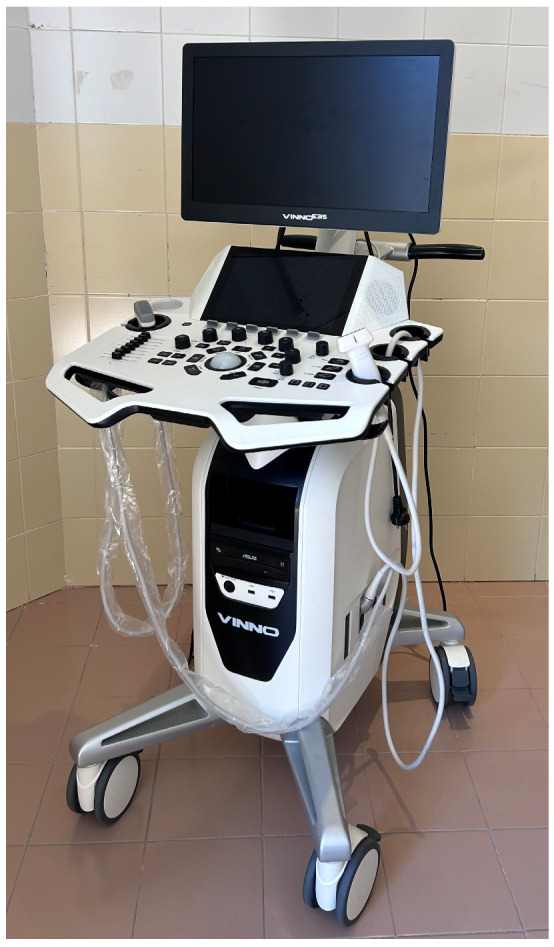
Vinno E35 ultrasound.

**Figure 2 healthcare-13-02484-f002:**
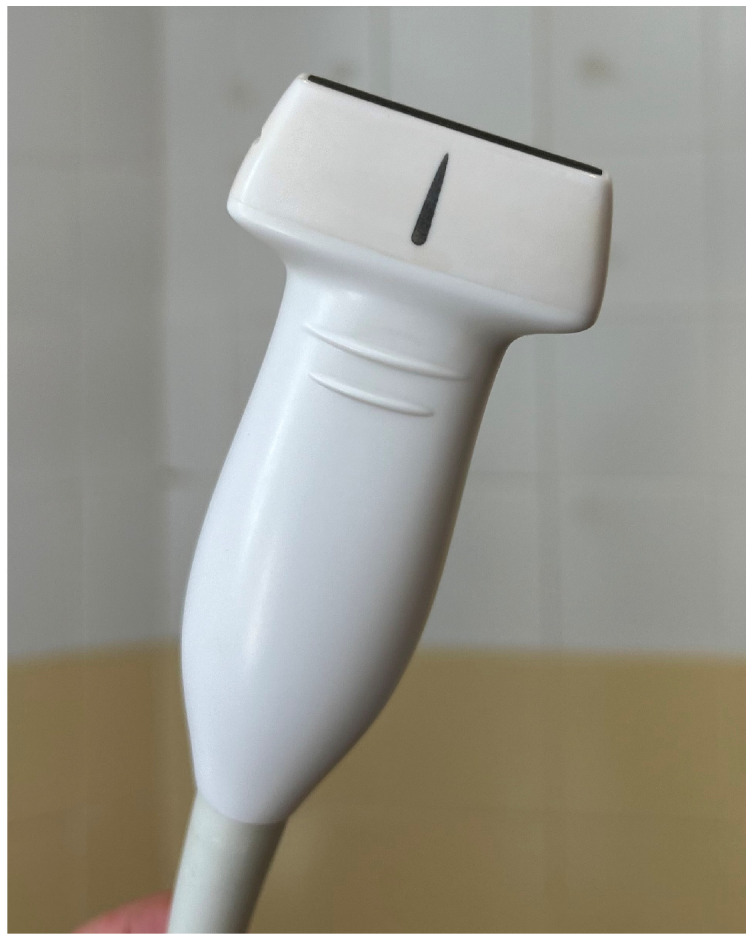
Linear transducer used in the study.

**Figure 3 healthcare-13-02484-f003:**
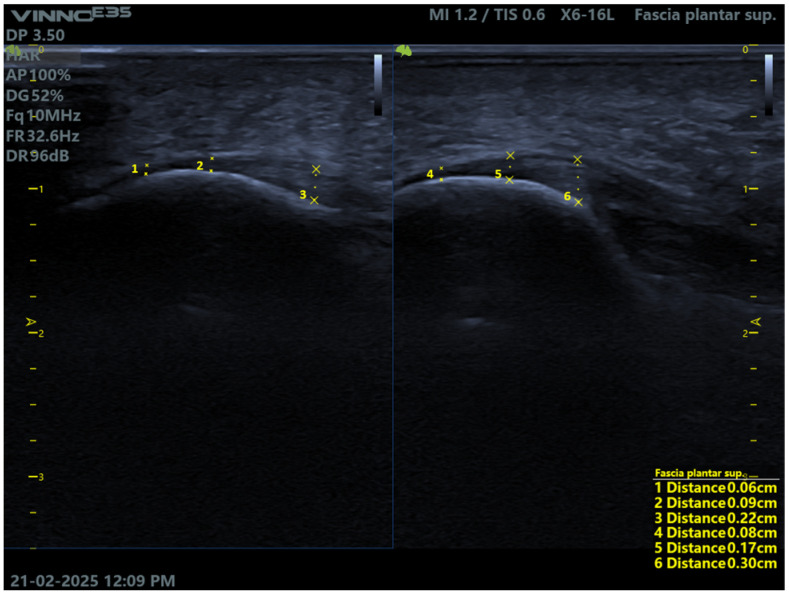
Comparison of plantar fascia measurements in the right and left foot by the novice technician.

**Table 1 healthcare-13-02484-t001:** Sample characteristics.

	N	Minimum	Maximum	Mean	Standard Deviation (SD)
Age (years)	60	20	32	23.85	2.81
Height (m)	60	1.52	1.90	1.67	0.09
BMI	60	18.36	33.30	24.77	3.63

**Table 2 healthcare-13-02484-t002:** Test results.

	M1_PI_N-M1_PD_N	M2_PI_N-M2_PD_N	M3_PI_N-M3_PD_N	M1_PI_C-M1_PD_C	M2_PI_C-M2_PD_C
Z	−0.80	−0.35	−0.84	−0.29	−0.21
Sig.	0.43	0.73	0.41	0.77	0.27

Notes: M = measurement, PI: left foot, N: novice, PD: right foot, C: qualified.

**Table 3 healthcare-13-02484-t003:** Average measurement time between expert and novice technician.

	Min	Max	Median	Mean	SD	*p*-Value
Novice Measurement Time	0′59″	4′52″	1′39″	1′51″	0′43″	**<0.001**
Expert Measurement Time	0′27″	1′42″	0′40″	0′44″	0′13″	

**Table 4 healthcare-13-02484-t004:** Wilcoxon test.

	Time Measurement of Novice Researcher—First Part	Time Measurement of Novice Researcher—First Part	*p*-Value
Mean	2′12″	1′31″	**<0.001**
N	30	30	
SD	0′52″	0′19″	

## Data Availability

Data is contained within the article.

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
