# Peer review of "Comparative Evaluation of Ultrasound Measurement of the Plantar Fascia Between Expert and Novice Technicians"

_healthcare, 2025, doi:10.3390/healthcare13192484_

Round 1
Reviewer 1 Report
Comments and Suggestions for Authors
Dear Authors,
I have thoroughly reviewed your research. While your research is valuable in terms of subject matter and scope, it requires numerous revisions. Therefore, I recommend that you make the revisions I will provide below in a detailed and optimal manner. I will re-evaluate your research after the necessary revisions have been made.
Introduction
Although this section is written in a way that moves from the general to the specific, the sentences used throughout the paragraphs are not sufficiently supported by the literature and there are inconsistencies. For example, in the second paragraph, many sentences need to be supported by specific references, but only one source is cited. This is also the case in all other paragraphs. Please revise all paragraphs with valid and accurate references. Explain all sentences requiring references with the necessary references.
Also, clearly state your main hypothesis after the purpose sentence.
Method
Please provide a detailed study design or experimental design for your research. Explain how the research was conducted in full detail.
Results
In this section, you mention that the data does not follow a normal distribution, but you present the results as means. Please also provide the median values. I also recommend adding Cohen's d effect sizes to see the effect levels.
Discussion
Please start this section with the major findings of your research.
Furthermore, in this section, you have only referred to your own results and those in the literature. The discussion seems to be a section containing more information. Instead, strengthen the discussion by supporting it with many more references. Provide your research results, other literature results (both supporting and unsupporting), and then provide more detailed and literature-supported results on the anatomical, physiological, and causal consequences of these results. You should do this for all paragraphs.
It will not be possible to evaluate the discussion in its current state without these revisions.
Best
Author Response
|
First of all, thank the reviewers for their contributions and suggestions, since, without a doubt, they are enriching for the article. Regarding the suggested changes, the following modifications have been made: Reviewer #1: The modifications proposed by reviewer 1 are included marked in red in the attached document. |
|
( ) |
(x) |
( ) |
- Introduction
Although this section is written in a way that moves from the general to the specific, the sentences used throughout the paragraphs are not sufficiently supported by the literature and there are inconsistencies. For example, in the second paragraph, many sentences need to be supported by specific references, but only one source is cited. This is also the case in all other paragraphs. Please revise all paragraphs with valid and accurate references. Explain all sentences requiring references with the necessary references. Also, clearly state your main hypothesis after the purpose sentence.
- Thank you for your input. The introduction has been completely revised. Lines 47-137
- Method
Please provide a detailed study design or experimental design for your research. Explain how the research was conducted in full detail.
- Thank you for your feedback. Lines 152-165. To avoid comparisons, the novice technician will first perform the measurements, who will be alone in the examination room with the patient. The novice technician will then leave the examination room and the experienced technician will enter. They will follow the same procedure, taking three measurements on each foot.
The procedure will begin with the patient lying supine on the examination table. The plantar fascia insertion area, the area of ​​interest for the corresponding measurements, will be palpated. Once we have located the area, we will apply conductive gel to the transducer and, positioning it longitudinally, we will proceed to find the fascia insertion area.
When we have located the desired area, we will freeze the image to perform the measurements more accurately. Using the distance program, we will mark the three measurements for each foot.
- Results
In this section, you mention that the data does not follow a normal distribution, but you present the results as means. Please also provide the median values. I also recommend adding Cohen's d effect sizes to see the effect levels.
- We have taken this into account, adding it to lines 243-252; 259-262.
- Discussion
Please start this section with the major findings of your research.
Thank you for your comment. We modified what you asked for and added in lines 265-309
Furthermore, in this section, you have only referred to your own results and those in the literature. The discussion seems to be a section containing more information. Instead, strengthen the discussion by supporting it with many more references. Provide your research results, other literature results (both supporting and unsupporting), and then provide more detailed and literature-supported results on the anatomical, physiological, and causal consequences of these results. You should do this for all paragraphs. It will not be possible to evaluate the discussion in its current state without these revisions.
- Thank you for your comment. We modified what you asked for and added lines 346-372.
- Thank you for your comment. We modified what you asked for and added lines 354-360.

Reviewer 2 Report
Comments and Suggestions for Authors
The manuscript clearly outlines the rationale for comparing technician experience levels during ultrasound measurements. However, the introduction could benefit from a more robust discussion regarding the potential clinical impact of operator variability. For instance, how would the findings translate to clinical practice when evaluating patients with pathology (e.g., plantar fasciitis) versus a healthy population?
It might be useful to explicitly state the hypothesis regarding measurement consistency and the learning curve of novice operators to better contextualize the study outcomes.
Although 60 healthy subjects are included, further discussion regarding the generalizability of these findings to a clinical population should be provided. The exclusion of patients with relevant pathologies may limit external validity.
The method of acquiring three independent measurements per foot for each operator is a strength. It is recommended to elaborate on whether the order of examinations (novice first, then expert) could have introduced any bias (e.g., through learning effects or probe repositioning) and, if so, how this potential bias was mitigated.
The use of non-parametric tests is appropriate given the distribution of the data. However, additional context regarding effect sizes would enhance the interpretation of the results. Also, please clarify the statistical approach used to compare the learning effect (e.g., inclusion criteria for dividing the sample into two phases).
The discussion accurately summarizes the findings and compares them with previous literature. It would be helpful to include a section addressing potential limitations (e.g., limited age range, healthy subject sample, potential order effects) and suggest ways these could be overcome in future studies.
Elaborating on the implications of the learning curve—how training protocols might be optimized based on these results—could enhance the discussion
Author Response
|
First of all, thank the reviewers for their contributions and suggestions, since, without a doubt, they are enriching for the article. Regarding the suggested changes, the following modifications have been made: |
|
( ) |
(x) |
( ) |
Reviewer #2.
The modifications proposed by reviewer 1 are included marked in red in the attached document.
- The manuscript clearly outlines the rationale for comparing technician experience levels during ultrasound measurements. However, the introduction could benefit from a more robust discussion regarding the potential clinical impact of operator variability. For instance, how would the findings translate to clinical practice when evaluating patients with pathology (e.g., plantar fasciitis) versus a healthy population?
It might be useful to explicitly state the hypothesis regarding measurement consistency and the learning curve of novice operators to better contextualize the study outcomes.
- The results of this study, obtained in a healthy population, show that there are no statistically significant differences between measurements performed by technicians with different levels of experience. However, it was observed that the expert technician was faster and more consistent in identifying anatomical structures, while the novice technician showed greater variability in recording times.
When extrapolating these findings to clinical practice, especially in patients with conditions such as plantar fasciitis, it is important to consider that the clinical situation differs substantially from that of the healthy population. First, the presence of thickening, inflammation, or other morphostructural alterations can make ultrasound identification difficult, increasing the risk of interobserver variability. Second, in a clinical context, small differences in measurements can have significant diagnostic and therapeutic implications, which places greater emphasis on the consistency and accuracy of the expert evaluator.
In this sense, although the results in healthy subjects suggest a low probability of systematic bias, in patients with pathology, the technician's experience could be a critical factor in ensuring the internal validity and reproducibility of the measurements. Furthermore, from the perspective of external validity, these findings highlight the need to consider the operator's training level when interpreting the clinical applicability of the results.
In conclusion, the absence of significant differences in healthy volunteers does not guarantee that the same findings will be replicated in a clinical population. On the contrary, it is plausible that in patients with plantar fasciitis, the evaluator's experience may be more relevant to the reliability of the measurement and, therefore, to the quality of clinical decision-making.
Thank you for your comment. We modified what you asked for and added lines 354-360
- Although 60 healthy subjects are included, further discussion regarding the generalizability of these findings to a clinical population should be provided. The exclusion of patients with relevant pathologies may limit external validity.
- Thank you very much for your contribution. This is an idea we have to continue advancing along this line of research on which we are currently working.
- Thank you for your feedback. Lines 354- 360
- The method of acquiring three independent measurements per foot for each operator is a strength. It is recommended to elaborate on whether the order of examinations (novice first, then expert) could have introduced any bias (e.g., through learning effects or probe repositioning) and, if so, how this potential bias was mitigated.
- Thank you for your feedback. Lines 152-161.
- The use of non-parametric tests is appropriate given the distribution of the data. However, additional context regarding effect sizes would enhance the interpretation of the results. Also, please clarify the statistical approach used to compare the learning effect (e.g., inclusion criteria for dividing the sample into two phases).
- We have taken this into account, adding it to lines 243-252; 259-262,
- The discussion accurately summarizes the findings and compares them with previous literature. It would be helpful to include a section addressing potential limitations (e.g., limited age range, healthy subject sample, potential order effects) and suggest ways these could be overcome in future studies. Elaborating on the implications of the learning curve—how training protocols might be optimized based on these results—could enhance the discussion
- We have taken this into account, adding it to lines 361-365

Reviewer 3 Report
Comments and Suggestions for Authors
The work might seem initially very interesting but then reading the arguments and conclusions reduces its academic thickness. Speaking of data precision and speed speed, it seems to me superfluous a research to demonstrate that an expert operator is better than a novice one on the study timing of the anatomical segment and that "in normality" a clinical examination must give the same results !!!! Nothing to complain about the structuring of the work, well articulated, clear and valid the statistical assessments that manage to support the numerous application variables thanks to the use of the normality tests of Kolmogorov-Smnnov and Shapiro-Wilk and the Wilcoxon test of the ranks as well as to the Conibm SPSS Statistics processing version 21 for iOS. Clear the tables for the demonstration of conclusions. Good choice of materials and methods in analyzing a population of men and women with a relatively young age but with anatomy now well structured, optimal the presence of the approval of the Ethical Committee "Bioethics and Biosicity Committee (Registration number: 53 // 2025)" of the University of Farmadura. In my opinion, the studies do not bring substantial novelties about the study of the footbed and some deductions seem too elementary even if they are studied.
Author Response
|
First of all, thank the reviewers for their contributions and suggestions, since, without a doubt, they are enriching for the article. Regarding the suggested changes, the following modifications or clarifications have been made: |
|
( ) |
(x) |
( ) |
Reviewer #3.
- The work might seem initially very interesting but then reading the arguments and conclusions reduces its academic thickness. Speaking of data precision and speed speed, it seems to me superfluous a research to demonstrate that an expert operator is better than a novice one on the study timing of the anatomical segment and that "in normality" a clinical examination must give the same results !!!! Nothing to complain about the structuring of the work, well articulated, clear and valid the statistical assessments that manage to support the numerous application variables thanks to the use of the normality tests of Kolmogorov-Smnnov and Shapiro-Wilk and the Wilcoxon test of the ranks as well as to the Conibm SPSS Statistics processing version 21 for iOS. Clear the tables for the demonstration of conclusions. Good choice of materials and methods in analyzing a population of men and women with a relatively young age but with anatomy now well structured, optimal the presence of the approval of the Ethical Committee "Bioethics and Biosicity Committee (Registration number: 53 // 2025)" of the University of Farmadura. In my opinion, the studies do not bring substantial novelties about the study of the footbed and some deductions seem too elementary even if they are studied.
- I greatly appreciate your observation. It is true that some conclusions may seem elementary at first glance; however, we believe the strength of the study lies precisely in addressing an area where scientific evidence is still very limited. The use of ultrasound for the analysis of the plantar fascia, and especially the comparison of variability between technicians with different levels of experience, represents an underexplored field of research.
To date, available studies on the plantar fascia have largely focused on describing clinical aspects of pathologies such as fasciitis, but there is little literature evaluating the reliability and consistency of ultrasound measurements across operators. In this sense, we believe the findings of this study provide added value, as they highlight the importance of technician experience in a context where diagnostic accuracy is critical to clinical practice.
In conclusion, although it may seem like a basic topic, we believe this is a pioneering study in musculoskeletal ultrasound applied to the plantar fascia, and that it can serve as a starting point for future, broader research in populations with pathology.

Round 2
Reviewer 1 Report
Comments and Suggestions for Authors
Manuscript is ready for publish
Author Response
Dear Reviewer,
Thank you very much for your positive evaluation and for recommending the publication of our manuscript. We truly appreciate the time and effort you dedicated to reviewing our work, as well as your constructive comments, which helped us improve the quality of the paper.
Your support is very encouraging, and we are pleased that the study has been considered a valuable contribution to the field.
Warm regards,
Carmen García-Gomariz
On behalf of all co-authors
Reviewer 3 Report
Comments and Suggestions for Authors
Regarding the changes made by the authors, I believe they are inadequate, as they attempted to salvage the work by modifying the final objectives of the paper, which I believe to be trivial. Scientific research is not necessary to demonstrate that experience helps us be precise in formulating scientific, academic, and clinical data. Therefore, in my opinion, the paper should be completely reworked, increasing the number of clinical cases and changing the purpose of the paper. I leave it to the editor to decide whether or not to publish a work that is certainly not excellent, even if well-structured from a research perspective, but with few clinical cases.
Author Response
First of all, thank the reviewer for their contributions and suggestions, since, without a doubt, they are enriching for the article. Regarding the suggested changes, the following modifications have been made:
Reviewer 3
- Regarding the changes made by the authors, I believe they are inadequate, as they attempted to salvage the work by modifying the final objectives of the paper, which I believe to be trivial. Scientific research is not necessary to demonstrate that experience helps us be precise in formulating scientific, academic, and clinical data. Therefore, in my opinion, the paper should be completely reworked, increasing the number of clinical cases and changing the purpose of the paper. I leave it to the editor to decide whether or not to publish a work that is certainly not excellent, even if well-structured from a research perspective, but with few clinical cases.
- Dear reviewer, we sincerely appreciate the time you took to review our manuscript and your valuable comments. We would like to respond respectfully to your comments, clarifying some key aspects of our approach.
We understand your concerns regarding the sample size and the objective of the study. However, we want to emphasize that this is a preliminary study involving measurements in 60 participants by both the novice and expert measurers, born precisely from the need to explore a line of research that has been little addressed until now. The sample used has already been fully collected and analyzed, and the main purpose has been to rigorously describe and analyze the findings, based on the existing literature. We believe this initial stage is essential to justify subsequent, larger studies with more ambitious objectives. We would like to highlight, positively, that despite having a sample of 60 participants, we obtained a moderate effect size, which is highly relevant and reinforces the results
Regarding the changes made after the initial review, we regret if they were not considered sufficient. Our goal was to maintain the internal consistency of the manuscript while addressing previous recommendations. However, we believe that substantially modifying the objectives or increasing the number of clinical cases at this stage would compromise the preliminary and exploratory nature of the study.
Despite the limitations mentioned, we believe that the results obtained, although modest, provide novel information within the field and could be of interest to other researchers wishing to develop more robust work on this topic. Therefore, we consider the article sufficiently worthy of publication as an initial contribution to the topic.
Of course, we respect the final editorial decision and remain at your disposal for any further clarification.
